# Engineering Microbial Consortia towards Bioremediation

Xianglong Li [1,2,†], Shanghua Wu [1,2,†], Yuzhu Dong [1,2], Haonan Fan [1,2], Zhihui Bai [1,2] and Xuliang Zhuang [1,2,*]

1    Key Laboratory of Environmental Biotechnology, Research Center for Eco-Environmental Sciences, Chinese Academy of Sciences, Beijing 100085, China; xlli_st@rcees.ac.cn (X.L.); shwu@rcees.ac.cn (S.W.); yzdong_st@rcees.ac.cn (Y.D.); hnfan_st@rcees.ac.cn (H.F.); zhbai@rcees.ac.cn (Z.B.)
2    College of Resources and Environment, University of Chinese Academy of Sciences, Beijing 100049, China
*    Correspondence: xlzhuang@rcees.ac.cn; Tel.: +86-010-6284-9193
†    Xianglong Li and Shanghua Wu contributed equally to this article.

**Abstract:** Bioremediation is a sustainable remediation technology as it utilizes microorganisms to convert hazardous compounds into their less toxic or non-toxic constituent elements. This technology has achieved some success in the past decades; however, factors involving microbial consortia, such as microbial assembly, functional interactions, and the role of member species, hinder its development. Microbial consortia may be engineered to reconfigure metabolic pathways and reprogram social interactions to get the desired function, thereby providing solutions to its inherent problems. The engineering of microbial consortia is commonly applied for the commercial production of biomolecules. However, in the field of bioremediation, the engineering of microbial consortia needs to be emphasized. In this review, we will discuss the molecular and ecological mechanisms of engineering microbial consortia with a particular focus on metabolic cross-feeding within species and the transfer of metabolites. We also discuss the advantages and limitations of top-down and bottom-up approaches of engineering microbial consortia and their applications in bioremediation.

**Keywords:** engineering microbial consortia; bioremediation; cross-feeding; top-down engineering; bottom-up engineering

## 1. Introduction

Microbes are ubiquitous organisms, found in air, soil, water, as well as animals, and plants [1]. They play vital roles in driving global biogeochemical cycles and have an immense impact on the survival, health, and development of mankind. A number of microbes have been isolated in laboratories that possess the ability to degrade organic pollutants and reduce or transform heavy metals [2]. However, the transforming efficiency of pollutants by a single species always declines when applied to in-site complex polluted sites [3]. Complex pollutants impose stress conditions on a single species and hinder their metabolism. In contrast, microbial consortia tend to show resistance and multifunctionality as varied species work together to efficiently utilize all forms of substrates [4,5].

Engineering microbial consortia may be an effective way to optimize the interaction within microorganisms and their environment and to ensure long-term stability. In the microbial communities, microbes compete for limited nutrients and consume metabolic products secreted by other species to gain fitness advantage [6]. It has been successfully applied in bioremediation of polluted sites, but also failed in some cases [7]. The metabolic interactions have a huge impact on the application of microbial consortia. The substances secreted by specific species can support or suppress the growth of other species, alter interactions between them and even influence the function of the whole community. Synthetic microbial consortia are defined as one that is created artificially by co-culturing of select (two or many) species under a (at least initially) well-defined media [8]. Unlike natural microbial consortia, it is feasible to reconfigure metabolic pathways and program social interactions of synthetic microbial consortia to obtain the desired function.

In this article, we focused on the molecular mechanisms of microbial consortia, particularly metabolic cross-feeding between species (Figure 1). We reviewed two main ways of engineering microbial consortia: top-down engineering and bottom-up engineering. Besides, we addressed important principles for engineering microbial consortia for the bioremediation of pollutants.

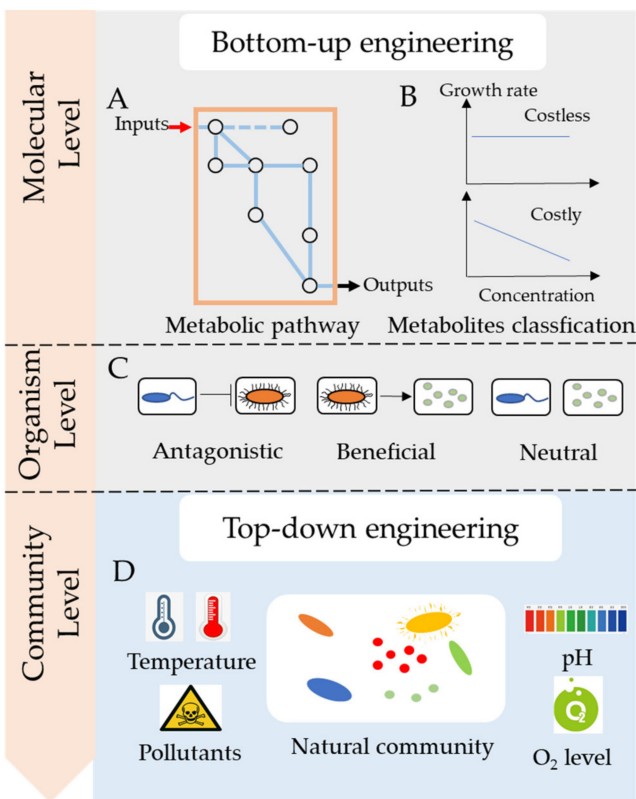

**Figure 1.** Approaches for engineering microbial consortia. There are two main approaches for engineering microbial consortia: bottom-up engineering and top-down engineering. Bottom-up engineering involves reconfiguring the metabolic pathway (**A**) and reprogramming social interactions (choose the species without reprogramming their metabolism). (**C**) The interactions between microbial consortia include antagonistic, neutral, or beneficial interactions (**B**) Metabolites are mainly divided into costless and costly depending on whether the secretion of metabolites causes a loss to fitness in monoculture (represented by growth rate). (**D**) Top-down engineering involves compelling a natural microbial community to perform desired functions by modifying environmental variables, such as temperature, pH, salt content, redox conditions, and carbon source.

## 2. Microbial Cross-Feeding in Microbial Consortia

Microbial cross-feedings are common in natural environments [9] and microbes frequently exchange substances [10]. Many studies have reported that microbes significantly benefit from this exchange of metabolites [11]. In order to improve microbial consortia's performance in bioremediation, it is essential to delineate the mechanisms of engineered microbial consortia on the molecular level by identifying metabolites and mechanisms of transfer.

### 2.1. Transfer of Metabolites between Microbes

Microbes commonly secrete metabolites across the membrane into the environment for utilization by other species. Several metabolites such as carbon and nitrogen resources, hydrogen ($H_2$), amino acids, vitamin, or growth factors may be exchanged between microbes [7]. These metabolites could be classified by molecule type, connection to metabolism, and fitness cost. Recently, more researches tended to focus on costly and

costless metabolites, as they have different influences on the stability of microbial consortia [6,12]. For example, Bidirectional of costly metabolites may promote the growth of each species [13]. In contrast, unidirectional exchange of costly metabolites harms the secretors, and may lead the microbial consortia to collapse [14]. Besides, the exchange of costless metabolites doesn't harm the secretors, instead benefits the receivers, thus, it causes no effect on the stability of microbial consortia. So, in this article, secreted metabolites are mainly classified into two categories, costly metabolites, and costless metabolites, depending on whether the secretion of metabolites causes a loss to the fitness of the organism.

### 2.1.1. Costly Metabolites

Costly metabolites usually benefit the receivers but harm the producers as the increased secretion leads to reduced fitness. Creating a bidirectional exchange of costly metabolites could benefit the microbial consortia, which may help construct a stable community to improve bioremediation. In a study, an *Escherichia coli* mutant unable to synthesize methionine was co-cultured with *Salmonella enterica* ser. *Typhimurium,* which was able to receive metabolites from *Escherichia coli* [13]. After several generations, the *Salmonella* strain underwent mutations to excrete methionine, thereby aiding *E. coli* growth. That is to say, *Salmonella* gained fitness by receiving nutrients from enhanced *E. coli* growth, which helped *Salmonella* to overcome the fitness cost of high methionine excretion. Therefore, the bidirectional exchange of costly metabolites contributes to a reciprocal nutrient exchange that is beneficial for the whole community. However, there was little research on determining whether the metabolites of pollutants were costly or not. It causes risks of disrupting the functional microbial consortia when unknown species are added.

### 2.1.2. Costless Metabolites

Microbes commonly secrete waste products with no fitness cost to the producer, that is, secretion does not alter the growth rate of the producer. It was proposed that costless metabolites can be a prominent driver of microbial interactions and influence the functions of microbial consortia [6]. The exchange of costless metabolites could offset competition for nutrients and yield stable specific partnerships, such as pollutants-degraders, which might help improve the bioremediation of pollutants. However, the cost of metabolite secretion and metabolic interactions of an organism may change in different environments [15] and are difficult to determine by experimental procedures. Using genome-scale models of metabolism, Pacheco et al. identified a large spectrum of costless metabolites [6].

### 2.2. Mechanisms of Metabolites Transfer and Metabolic Interactions

The ecological and evolutionary factors influencing mechanisms of metabolite transfer have been comprehensively reviewed by Glen D'Souza et al. [16]. There are two main modes of metabolites exchange: contact-independent mechanisms including passive diffusion, active transport, and vesicle-mediated transport, and contact-dependent mechanisms including vesicle chains, nanotubes, flagella-like filaments, and cell-cell contact. The stability and efficiency of cross-feeding are mainly affected by the compositions of microbial consortia and environmental factors like pH, temperature, and nutrient availability [17–19]. In a study, the addition of cross-fed nutrients changed the cross-feeding interaction by releasing one bacterial strain from its dependence on the other [20]. Additionally, spatial structures also have an impact on cross-feeding interactions. One model showed that resource diffusion and microbe dispersal determines the composition dynamics of a microbial consortium of three bacterial strains [21]. Cross-feeders could outcompete cheaters if microbe dispersal is low but resources are shared widely. While cheaters replaced cross-feeders and eventually leading to the collapses of the colony if microbe dispersal is high and resources are shared widely. It provided strategies to keep microbial consortia stable during bioremediation.

The interactions within microbial consortia include antagonistic, neutral, or beneficial interactions [16]. When microbes associate with different species, the interactions are

dynamic and may be influenced by environmental conditions. It has been reported the concentration of nutrients could alert the interactions between different species [22,23]. For example, ammonia concentration could alter the interactions within synthetic communities from collaborating to competing [24]. However, limited knowledge of changes of interaction within microbial consortia under different pollution hinders the application of microbial consortia to restore contaminated environments.

## 3. How to Engineer Microbial Consortia towards Bioremediation

The engineering of microbial consortia is a complex task involving the assembly of different genera and species of microbes. There are two main approaches and several important principles of engineering microbial consortia towards bioremediation.

### 3.1. Top-Down Engineering

Top-down engineering utilizes a natural microbial community to perform desired functions by modifying environmental variables, such as temperature, pH, mean incubation times, salt content or redox conditions, and carbon resource [25]. It provided a convenient way for bioremediation. For example, a study showed that adding electron donors contributed to the growth of microbial consortia and helped improve the degradation of toxic chlorinated contaminants [26]. In top-down engineering, microbial consortia are considered an integral unit with defined functions without considering species specificity and metabolic pathways. In this approach, microbial consortia are self-assembled and the growth of key species is promoted by using reactor engineering or biostimulation. In an anoxic pond of wastewater-treatment plants, denitrifying bacteria were enriched in the microbial community to turn nitrate to $N_2$.

In top-down engineering, it's crucial to manipulate the whole microbial consortia. Modeling of microbial consortia contributed to this by predicting whether the desired function will be obtained. Physicochemical conditions (pH, temperature, oxygen content), abiotic, and biotic processes all have influences on biological processes to some extent. By capturing specific stoichiometric parameters and kinetic parameters (substrate uptake rate) of microbial consortia, the function of the community can be predicted. Besides, modeling could help improve the efficiencies of removing pollutants and control the costs of bioremediation.

Top-down engineering is a powerful conventional approach involving environmental manipulation and has been proved successful. However, its limitation lies in the lack of accurate control over the microbial consortia. Top-down engineering overlooks the complex interconnected metabolism and functional interactions in microbial consortia, thus limiting system optimization.

Eliminating Stress Arising from Complex Pollutants

Many polluted sites are co-contaminated with organic and metal pollutants which cause a combined stress effect to microbes [3]. Biodegradation of the organic component can be influenced by metal toxicity by impacting enzymes directly involved in bioremediation or general metabolic enzymes [27]. In addition, metals, like Cr(VI), may cause damage to cytomembrane or DNA [28]. Similarly, the presence of organic pollutants also imposed stress conditions on microorganisms. PAHs such as PHE, PYE, are severely toxic to cell membranes resulting in membrane dysfunction or destabilization [29]. Additionally, PAHs may alter the accumulation of metals and enhance metal-derived reactive oxygen species (ROS) [30]. So, eliminating stress arising from complex pollutants is important in the progress of top-down engineering.

Adding stimulating factors, such as carbon, nitrogen, phosphorus resources, and surfactants, has been proved to help microbial consortia overcome this stress [31]. Some stimulating factors have the ability to promote the formation of biofilms. It has been reported that the formation of biofilm increased the tolerance of microbial consortia. For instance, when biofilms of *Zymomonas mobilis* were exposed to benzaldehyde, the metabolic

activity of cells was reduced by only 50%, as compared to total inhibition in planktonic cells [32]. However, more researches needed to be done to address the mechanism of increasing tolerance through top-down engineering.

### 3.2. Bottom-Up Engineering

Bottom-up engineering is based on the metabolism and interactions between species. In the past, it was a challenge to predict and precisely manage microbial consortia; however, with the development of multi-omics and automation technology, many researchers have found ways to manipulate metabolic networks and microbial interactions [26,33,34].

Metabolism determines the nutrients that a species consumes and the metabolites that are excreted into the environment. There are several pathway models for the molecular mechanisms, such as glycolysis, the Kreb's cycle, and glutaminolysis. Constraint-based methods, such as flux balance analysis, were also used for bottom-up engineering. Based on metabolic fluxes and interacting networks, it is possible to predict the desired output. Generally, the genome sequences of the strains in microbial consortia are needed to reconstruct the metabolic pathways and quantitative models are used to investigate the dynamics of the consortia [34]. Designing microbial consortia with defined social interactions is also an important part of bottom-up engineering. Microbial social interactions, such as competition and cooperation, are common in microbial communities and are essential in specifying ecosystem dynamics [35]. A model based on social-interaction programming can successfully predict the behaviors and dynamics of a community comprising of up to four species [36].

Based on bottom-up engineering, a microbial consortium with specific functions can be constructed and optimized. This approach has advantages in the bioremediation of complex pollutants. However, species with inaccurate and/or incomplete data regarding metabolic networks, genes, or proteins with unknown functions cause a major challenge in the engineering of microbial consortia.

### 3.2.1. Making Division of Labor in Metabolic Pathways

The metabolic pathways of some pollutants are complex and degradation of pollutants involves several steps. Microbes have different metabolic mechanisms and resource preferences, which means different costs for metabolite production. So, it might not be energy-efficient for one species to independently perform this whole progress. Besides, certain intermediates of pollutant degradation are more toxic than the original compound and inhibit degradation, such as the metabolic intermediates of trichloroethylene (TCE) [37]. It's necessary to make a division of labor in metabolic pathways for degrading some pollutants.

Ensuring division of labor in metabolism promotes bioremediation as one species may have a comparative advantage over other strains in a specific step in the degradation of pollutants. In a synthetic pyrene-degrading microbial consortium, only two *Mycobacterium* strains contribute to the initial steps of pyrene degradation. *Novosphingobium* and *Ochrobactrum* did not degrade pyrene; however, these bacteria could efficiently degrade the intermediates of pyrene [38]. Although *Mycobacterium* strains possessed a complete gene set for pyrene degradation, they did not metabolize pyrene. *Novosphingobium* and *Ochrobactrum* degrade phthalate or protocatechuate more efficiently than *Mycobacterium* strains, so they have more comparative advantages and contribute to the degradation of intermediates of pyrene. In a synthetic atrazine-degrading microbial consortium consisting of four members, the growth and degradation rate of the microbial consortium is more efficient than that of a single species [39]. Although only one species is able to directly mineralize atrazine, the other three utilize intermediates during the degradation process. Each species has unique advantages, and the division of labor in metabolism helps microbial consortia degrade pollutants more economically and effectively (Figure 2).

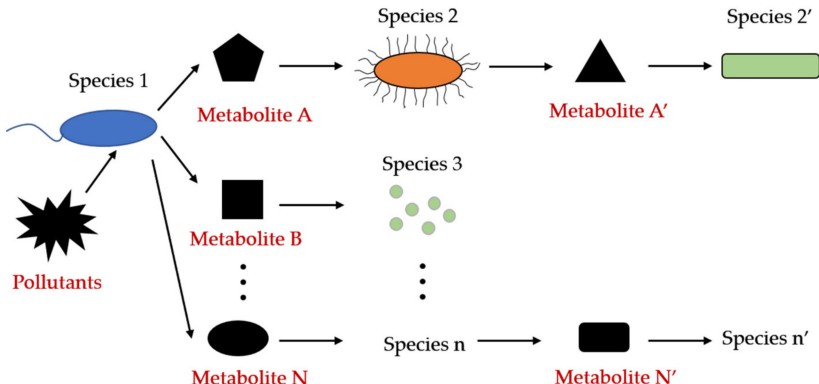

**Figure 2.** Division of labor in a metabolic pathway.

### 3.2.2. Keeping Multifunctionality of Microbial Communities

In the natural environment, microbes tend to assemble and form a multifunctional community. Research has shown that the diversity of a microbial consortium contributes to its stability [40]. This is because microbial communities with diverse functions are robust owing to enhanced access to nutrients and support from core partner species, allowing them to be unaffected by environmental disturbances or stresses and have improved functions. For example, in a microbial consortium comprising of *Salmonella enterica ser. Typhimurium* and *Escherichia coli* mutant, *Salmonella* is unable to directly utilize lactose, the only carbon resource; however, *Escherichia coli* can metabolize lactose and provide *Salmonella* with costless metabolic byproducts [12].

The core function of synthetic microbial consortia in the bioremediation of pollutants is the degradation of hazardous substances. Therefore, degraders are indispensable in a synthetic microbial consortium. However, there are still other important partner species with varied functions in the consortia that could contribute to the improvement of bioremediation (Figure 3). When dealing with low water-soluble organic pollutants, the surfactant-producing function of microbial consortia needed to be considered. When the nutrient of polluted sites was poor, adding nitrogen fixers and carbon producers was important to provided degraders with enough nutrients for bioremediation. For instance, a *Bacillus* strain, part of a pyrene-degrading consortium, is unable to degrade pyrene, but can enhance the bioavailability of pyrene by producing biosurfactant [38]. A polycyclic aromatic hydrocarbons (PAHs)-degrading microbial consortium consisting of *Pseudomonas* and *Actinobacteria* strains also show emulsifying activities in the presence of PAHs, which notably helps the solubilization of PAHs during biodegradation [41].

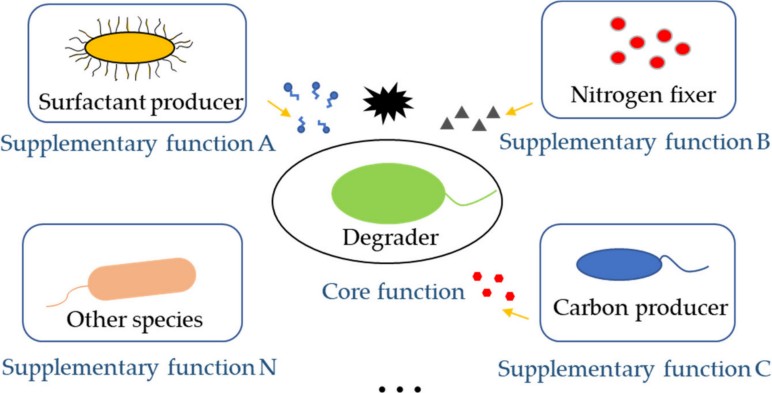

**Figure 3.** Multifunctionality in a microbial consortium. Degradation of pollutants is a core function of a synthetic microbial consortium. Supplementary functions, such as producing surfactant, nitrogen, or carbon resources, help the degraders to achieve the core function.

## 4. Engineering Microbial Consortia Promotes Bioremediation

In recent years, engineering microbial consortia have been used to study the degradation of organic pollutants or the removal of heavy metals. Certain synthetic microbial consortia have exhibited promising potential for bioremediation of polluted sites (Table 1).

**Table 1.** Cases of engineering microbial consortia towards bioremediation.

| Pollutants | Microorganism | Bioremediation Efficiency | References |
|---|---|---|---|
| Pyrene | (*Mycobacterium spp*. PO1 and PO2, *Novosphin-gobium pentaromativorans* PY1, *Ochrobactrum* sp. PW1, and *Bacillus* sp. FW1 | Three-fold higher degradation rate for pyrene than the individual degrader. | [38] |
| Atrazine | *Arthrobacter* sp. DNS10, *Bacillus subtilis* DNS4 and *Variovorax* sp. DNS12, *Arthrobacter* sp. DNS9 | Removed 100% of atrazine at initial concentration of 100 mg/L, faster than single species. | [39] |
| PAHs | *Rhodococcus* sp., *Acinetobacter* sp., and *Pseudomonas* sp. | 100% degradation of Fl and Phe in sediment-free liquid medium after 4 weeks of growth. | [42] |
| Cr(VI) | *Streptomyces* sp. A5, A11, M7, MC1 | Removed 86% of Cr(VI) at initial concentration of 50 mg/kg in soil. | [43] |
| Lindane | | Removed 46% of lindane at initial concentration of 25 mg/kg in soil. | |
| Cd | *Bacillus* sp. strain H9, *Ralstonia eutropha* JMP134 | Removed 42% of phenanthrene at initial concentration of 24 mg/L. | [44] |
| 2,4-D | | Removed 100% of 2,4-D at initial concentration of 500 mg/L. | |

### 4.1. Organic Pollutants

Organic pollutants such as high molecular weight polycyclic aromatic hydrocarbons (PAHs) are not efficiently degraded by a single strain [35,45]. However, a synthetic microbial consortium may perform better degradation under the approach of bottom-up engineering. A microbial consortium comprising of five culturable bacteria shows a three-fold higher degradation rate for pyrene than the single bacterial strain [38]. The biodegradation of pyrene was enhanced because of the cooperation between different species. Pyrene was initially degraded by *Mycobacterium*, while *Novosphingobium pentaromativorans* PY1, *Bacillus* sp. FW1, and *Ochrobactrum* sp. PW1 degraded the intermediates of pyrene. Besides, a biosurfactant was produced by *Bacillus* sp. FW1, which enhanced the dissolution of pyrene. The approach of top-down engineering also showed potential. In another study [42], mixed PAHs of fluorene (Fl), phenanthrene (Phe), and pyrene were effectively degraded by a synthetic microbial consortium consisting of three bacterial strains. The degradation rate of Fl and Phe was up to 100% after 4 weeks. Engineering microbial consortia exhibit a promising potential in the bioremediation of organic pollutants. However, the common approach is top-down engineering and microbial consortia are enriched from the in-situ environment. This is a time-consuming procedure and synthetic microbial consortia cannot be effectively applied in different sites. In the future, engineering microbial consortia based on bottom-up engineering requires more attention.

### 4.2. Heavy Metals

Engineering microbial consortia is an effective way for the removal of heavy metals. In a study [46], microbial consortia enhance the bioremediation of acid mine drainage, as biofilms formed by heterotrophic acidophiles decrease the dissolution rate of heavy metals. Biofilms play vital roles in the removal of heavy metals as they protect microbial consortia from diverse environmental stresses. Extracellular polymeric substances, such as polysaccharides, in biofilms, can easily bind to heavy metal ions [47]. However, the

interactions of microbial consortia in biofilms are still not clear and need more research. Furthermore, microbial consortia are found to be more effective for Cr(VI) bioremediation, such as sulfate-reducing microbial consortia reduces the toxicity of Cr(VI) by reducing it to Cr(III) [48]. The interactions within microbial consortia may promote the growth of Cr(VI)-reducing bacteria, and then contribute to Cr(VI) bioremediation [49]. Though engineering microbial consortia succeed in some applications, the costs analysis needed to be considered in the future.

*4.3. Complex Pollution*

Recent empirical studies have shown that bacteria significantly benefit from trading metabolites with others, and cross-feeding within microbial consortia augment their ability to survive in complex polluted environments [45]. In a synthetic microbial consortium, *Escherichia coli* ATCC 33456 and *Pseudomonas putida* DMP-1 contribute to the simultaneous degradation of phenol and reduction of Cr(VI) [50]. Polti et al. also constructed a microbial consortium including *Streptomyces* sp. M7, MC1, A5, and *Amycolatopsis tucumanensis* AB0, and the synergistic removal efficiency of Cr(VI) and lindane was up to 69.5% and 54.7%, respectively [43]. Microbial consortia are robust and resist stress by complex contaminants. Recently, interest in synthetic microbial consortia that can perform complex functions, unlike a single community, is gaining momentum.

## 5. Conclusions

The advancement of biotechnology helps in engineering microbial consortia with desired functions associated with bioremediation. Top-down engineering compels a natural microbial community to perform the desired functions by modifying environmental variables, while bottom-up engineering reconfigures the metabolic pathway and reprograms social interactions among microbes. Both approaches could be used for engineering microbial consortia towards bioremediation. Top-down engineering is effective and costless when dealing with simple and easily degraded pollutants. While bottom-up engineering has advantages in degrading complex pollutants. Besides, in the progress of engineering, principles such as making division of labor in metabolic pathways, keeping multifunctionality, and eliminating stress need to be considered. In the future, more works need to be focused on metabolic interactions between species to make accurate control over the microbial consortia to obtain desired functions and stable effects. Although there are many limitations in the application of engineered microbial consortia, it is still a promising approach for bioremediation.

**Author Contributions:** Conceptualization, X.L. and S.W.; validation, Z.B. and X.Z.; formal analysis, X.L.; investigation, X.L.; resources, X.L.; data curation, X.L.; writing—original draft preparation, X.L.; writing—review and editing, S.W., H.F. and Y.D.; visualization, X.L.; supervision, X.Z. and Z.B.; project administration, X.Z.; funding acquisition, X.Z. All authors have read and agreed to the published version of the manuscript.

**Funding:** Please add: This research was funded by the National Key R&D Program of China (Nos. 2019YFC1805803), Science and Technology Service Network Initiative of the Chinese Academy of Sciences (Grant KFJ-STS-ZDTP-064), the National Natural Science Foundation of China (Nos. 91951108, 41907273 and 31670507) and the Strategic Priority Research Program of Chinese Academy of Sciences (Grant No. XDA23010400).

**Conflicts of Interest:** The authors declare no conflict of interest.

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
