# Peer review of "Engineering Microbial Consortia towards Bioremediation"

_water, doi:10.3390/w13202928_

Round 1
Reviewer 1 Report
This review aims to provide a “how-to” for engineering microbial consortia specifically for bioremediation. It joins other recent reviews on similar topics, from the more general (https://www.nature.com/articles/ismej201626, https://microbialcellfactories.biomedcentral.com/articles/10.1186/s12934-019-1083-3, https://www.sciencedirect.com/science/article/abs/pii/S0734975019302009, Common principles and best practices for engineering microbiomes | Nature Reviews Microbiology ) to more specifically about bioremediation (https://www.tandfonline.com/doi/abs/10.1080/07388551.2020.1853032, https://www.sciencedirect.com/science/article/abs/pii/S2352186419306893 ).
This review shows potential, but at the moment is not particularly different in scope than other recent reviews. I think what could set it apart would be pushing hard on the “how-to” aspect, and / or focusing more on successes and failures in bioremediation (while arguing why they succeeded / failed), rather than spending a large section on crossfeeding or division of labor, which have been extensively covered elsewhere. Right now, the bioremediation aspect of the review is a scant 3 paragraphs in one page and is more descriptive than prescriptive (section 4). In my opinion, if the goal of the review is to discuss how to engineer communities for bioremediation, then it would be much improved by expanding section 4 dramatically, and reducing or removing other sections.
A major criticism is that many aspects of consortia are stated as facts, but without stating why they are relevant (evolutionarily, ecologically, or for bioremediation). For example, in lines 114-116 it is stated that “resource diffusion and microbe dispersal determine the composition” of a community. Why is this so? How should an engineer think about this? How might it relate to a bioremediation site? Related, it is not always clear what the relevance of some entire sections are to engineering of bioremediation communities. For instance, section 2.1 spends 4 paragraphs talking about how excreted metabolites can be costly or costless, without connecting this back to engineering or bioremediation. Additionally, top-down and bottom-up engineering are introduced, but no clear guidance is given on when one or the other strategy should be used. There are some good ideas in here, but considerable effort needs to be made to tie them back to the goal of the review.
Other:
- The manuscript switches from past tense to present tense periodically and inconsistently
- Line 33: cooperate has a specific ecological meaning, in which the cooperators are incurring a personal cost to reap a (hopefully) larger benefit. I would remove “cooperate to”
- Line 39: I’m also hopefully that consortia engineering will be an effective way to optimize interactions, but at this moment I think it is premature to claim that it “is” an effective way. Perhaps soften to “may be”
- Line 40: what is the “microbial world”?
- Line 42: unclear phrasing, hard to tell if the authors are arguing that crossfeeding is related to persistence in a “particular niche” or if the two clauses of the sentence should be interpreted independently
- Line 49: please cite the successes and failures
- Line 59: arguably bottom-up could be as simple as choosing the species without reprogramming their metabolism
- Line 62: important to clarify that the fitness cost is in monoculture; presumably in co-culture it improves fitness through the indirect mutualistic interaction
- Section 2 introduction: It isn’t clear from this introduction why this section is critical for understanding engineering of bioremediation communities. Could division of labor or stable population dynamics (or something else) be discussed briefly?
- Lines 78-80: costly vs. costless is one way to categorize secreted metabolites, but there are many others: type of molecule (amino acid / sugar / organic acid etc.), connection to metabolism (primary metabolic product vs. secondary metabolic product); effect on secondary species (benefit vs. toxin) etc. A good argument is needed to specify why costly vs. costless is the most important consideration for bioremediation communities.
- Line 82: “harm the producer” is unspecific, as it could include both a fitness cost from not using the metabolite (which is what the authors mean) or that the secreted metabolite is toxic to the secreter.
- Line 83: “mutation is the main reason for the secretion of costly metabolites” is quite vague. Couldn’t that be stated for any phenotype?
- Line 85: “public goods” are any publicly available good and don’t need to be costly
- Lines 85-86: this section of the review is on cross-feeding, but none of these public goods are crossfed (with the potential exception of siderophores)
- Lines87-89: an important part of that study was that Salmonella obligately depended on E coli—it was hypothesized that the mutual obligate dependence was required for the evolution of costly exchange
- Line 102: that Pacheco paper is great, but it is too strong to claim from a computational study that something has been proven
- Line 110: what is meant by “characteristics”
- Line 116: could this be expanded to say why or how diffusion mattered?
- Line 124: it is a strong claim that it is not energy-efficient for one species to perform a whole degradation pathway; this is still a hot area of active research
- Figure 2 does not show how or why division of labor can be more efficient than a single species using the whole pathway
- Line 150: define “stability”
- Line 151: what is the evidence that diverse functionality is robust (definition?) specifically because of enhanced access to nutrients and partner support, and how does diverse functionality lead to partner support?
- lines 162-167 is fascinating: it shows how engineering strategies should consider not just degradation of the pollutant, but also accessory functions which may indirectly speed the process. This is also the point of figure 3, which is interesting and could be greatly expanded upon. Could the authors find a way to make this a general strategy? When should or should it not be used?
- Line 183: Wouldn’t a top-down approach not require identifying all pollutants? Isn’t this only true for a bottom-up strategy?
- Line 184: why has adding these factors helped overcome stress? How would an engineer know which to use and when?
- Line 204: isn’t adding electron donors bottom-up?
- Lines 207-211: this is confusing, because I was under the impression that top-down engineering can let the “how” be a black box, in which the engineer doesn’t care how the function is achieved, as long as it is achieved
Author Response
Thanks for your useful comments.
Engineering microbial consortia is supposed to be a promising approach in biotechnology, many works about cross-feedings have been down, but it is rarely used for bioremediation, especially for those complex pollutants. In this article, we aimed to review useful methods for engineering microbial consortia in the degradation of complex pollutants. So, we firstly discussed the molecular and ecological mechanisms of engineering microbial consortia in section 2, because these mechanisms might have a huge effect on the interactions between microbial consortia and the efficiency of bioremediation. Then, we paid attention to strategies on ‘how to engineer communities for bioremediation’, and addressed important principles of engineering microbial consortia specifically for bioremediation.
Also, in the revised manuscript, we added the guidance on when one or the other strategy should be used following your suggestions.
Other:
The manuscript switches from past tense to present tense periodically and inconsistently
Line 33: cooperate has a specific ecological meaning, in which the cooperators are incurring a personal cost to reap a (hopefully) larger benefit. I would remove “cooperate to”
Response: We have changed ‘cooperate to’ to ‘work together to’ (please see line 33).
Line 39: I’m also hopefully that consortia engineering will be an effective way to optimize interactions, but at this moment I think it is premature to claim that it “is” an effective way. Perhaps soften to “may be”
Response: We have changed ‘is’ to ‘may be’ (please see line 39).
Line 40: what is the “microbial world”?
Response: We have changed “microbial world” to “microbial communities” (please see line 41).
Line 42: unclear phrasing, hard to tell if the authors are arguing that crossfeeding is related to persistence in a “particular niche” or if the two clauses of the sentence should be interpreted independently
Response: We have rewritten this sentence (please see line 42).
Line 49: please cite the successes and failures
Response: We have added references (please see line 43).
Line 59: arguably bottom-up could be as simple as choosing the species without reprogramming their metabolism
Response: Yes, we have added your comment into our revised manuscript (please see line 64).
Line 62: important to clarify that the fitness cost is in monoculture; presumably in co-culture it improves fitness through the indirect mutualistic interaction
Response: We have added ‘in monoculture’ to make this sentence clearer (please see line 67).
Section 2 introduction: It isn’t clear from this introduction why this section is critical for understanding engineering of bioremediation communities. Could division of labor or stable population dynamics (or something else) be discussed briefly?
Response: We have rewritten this section to highlight the importance for understanding engineering of bioremediation communities (please see section 2.1).
Lines 78-80: costly vs. costless is one way to categorize secreted metabolites, but there are many others: type of molecule (amino acid / sugar / organic acid etc.), connection to metabolism (primary metabolic product vs. secondary metabolic product); effect on secondary species (benefit vs. toxin) etc. A good argument is needed to specify why costly vs. costless is the most important consideration for bioremediation communities.
Response: We have added the explanations in this paragraph (please see line 83).
Line 82: “harm the producer” is unspecific, as it could include both a fitness cost from not using the metabolite (which is what the authors mean) or that the secreted metabolite is toxic to the secreter.
Response: Thanks for your suggestions. We cited the definition by Alan R. in ‘Costless metabolic secretions as drivers of interspecies interactions in microbial ecosystems’(2019), And secreted metabolites are mainly classified into two categories, costly metabolites and costless metabolites, depending on whether the secretion of metabolites causes a loss to the fitness of the organism. The secreted metabolites which are toxic to the secreter might belong to costless metabolites. The costly metabolites certainly harm the producer as they caused a loss to the fitness of the organism (please see line 90).
Line 83: “mutation is the main reason for the secretion of costly metabolites” is quite vague. Couldn’t that be stated for any phenotype?
Response: We have removed this sentence (please see line 92).
Line 85: “public goods” are any publicly available good and don’t need to be costly
Response: We have removed this sentence (please see line 93).
Lines 85-86: this section of the review is on cross-feeding, but none of these public goods are crossfed (with the potential exception of siderophores)
Response: Thanks for your suggestions. These public goods all have potential for cross-feeding, as they could be utilized or degraded by other species (please see line 88).
Lines87-89: an important part of that study was that Salmonella obligately depended on E coli—it was hypothesized that the mutual obligate dependence was required for the evolution of costly exchange
Response: Thanks for your suggestions. Creating bidirectional exchange of costly metabolites could benefit the microbial consortia, which may help construct a stable community to improve bioremediation.
Line 102: that Pacheco paper is great, but it is too strong to claim from a computational study that something has been proven
Response: We have changed ‘proved’ to ‘proposed’ (please see line 109).
Line 110: what is meant by “characteristics”
Response: We have changed it to ‘compositions’ (please see line 128).
Line 116: could this be expanded to say why or how diffusion mattered?
Response: We have added more explanations (please see line 134).
Line 124: it is a strong claim that it is not energy-efficient for one species to perform a whole degradation pathway; this is still a hot area of active research
Response: We have changed it to ‘might not be’ (please see line 146).
Figure 2 does not show how or why division of labor can be more efficient than a single species using the whole pathway
Response: We have revised the Fig. 2 (please see line 169).
Line 150: define “stability”
Response: It means the capacity of resisting disturbance for microbial consortia.
Line 151: what is the evidence that diverse functionality is robust (definition?) specifically because of enhanced access to nutrients and partner support, and how does diverse functionality lead to partner support?
Response: We have set examples after this sentence (please see line 181).
lines 162-167 is fascinating: it shows how engineering strategies should consider not just degradation of the pollutant, but also accessory functions which may indirectly speed the process. This is also the point of figure 3, which is interesting and could be greatly expanded upon. Could the authors find a way to make this a general strategy? When should or should it not be used?
Response: We have expanded this part (please see line 188).
Line 183: Wouldn’t a top-down approach not require identifying all pollutants? Isn’t this only true for a bottom-up strategy?
Response: Thanks for your suggestions, we have removed this sentence (please see line 212).
Line 184: why has adding these factors helped overcome stress? How would an engineer know which to use and when?
Response: We have added related references (please see line 215).
Line 204: isn’t adding electron donors bottom-up?
Response: Thanks for your suggestion, in fact we think it might belong to both approaches. Top-down engineering utilizes a natural microbial community to perform expected functions by modifying environmental variables. Adding electron donors is an approach to modify environmental variable.
Lines 207-211: this is confusing, because I was under the impression that top-down engineering can let the “how” be a black box, in which the engineer doesn’t care how the function is achieved, as long as it is achieved
Response: Thanks for your suggestions. In top-down engineering, microbial consortia are considered an integral unit. Approaches, such as adding electron donors, modulating pH or temperature, could be used to obtain the expected functions. Besides, modeling could also be used to predict whether a desired function will be obtained.
Reviewer 2 Report
This paper, entitled Engineering microbial consortia towards bioremediation, is a scholarly work and can increase knowledge on this topic. The content is relevant to Water and should generate new knowledge. The manuscript is quite well written and well related to existing literature. The authors provide an interesting paper dealing with engineering microbial consortia in the domain of bioremediation. The authors discuss the advantages and limitations of top-down and bottom-up approaches of engineering microbial consortia and its applications in bioremediation.
From my point of view, this paper is very interesting but there's some points missing. What are the limitations of such approach in terms of transfer at real scale? What is the viability of such approach in terms of costs? Please provide costs analysis or costs considerations by giving costs vs benefits discussion. Is it realistic and viable for all application? I will appreciate if authors could provide or mention the TRL of such approach.
I recommend the following decision due to these main comments: ACCEPT AFTER MINOR REVISION.
Author Response
Many thanks for your good words on our manuscript. We really appreciate that our manuscript brings your interests.
Engineering of microbial consortia is commonly applied for commercial production of biomolecules. It might reach TRL9. However, in the field of bioremediation, this approach is still immature. As for now, there are still some microbial consortia could not get a stable and outstanding results of bioremediation, which hinder its application to restore contaminated environments. As we addressed in the manuscript, limited knowledge of the assembly and interaction of bacteria hinders the efficiency of bioremediation. So, we mainly reviewed approaches of effectively engineering microbial consortia to remove targeted pollutants. Its cost and benefits also vary largely when dealing with different pollutants. We have briefly added the discussions of costs analysis in section 4.2. We will make fully costs analysis when we deal with specific pollutant in the future.
Reviewer 3 Report
- In this article, the molecular mechanisms of microbial consortia were discussed. Although important principles for engineering microbial consortia for bioremediation of pollutants were addressed, however some important enzyme reactions were not introduced in this article. Please tabulate the main enzyme reactions in the molecular mechanisms of microbial consortia.
- In the past, several pathway models for the molecular mechanisms of microbial consortia were proposed. Please add these pathway models and introduce them briefly.
- Please discuss the metabolic flux analysis for the molecular mechanisms of microbial consortia.
- Please describe the novelty and innovation of this paper.
- Please describe the applicability.
- Please discuss the limitations of practical application.
- Please describe how to apply the results in environmental management.
Author Response
Thanks for your useful suggestions.
In this article, the molecular mechanisms of microbial consortia were discussed. Although important principles for engineering microbial consortia for bioremediation of pollutants were addressed, however some important enzyme reactions were not introduced in this article. Please tabulate the main enzyme reactions in the molecular mechanisms of microbial consortia.
Response: In this article, we paid attention to microbial consortia and metabolic interactions between them. And we also added introduction of important enzymes in section 2.1.1, which may have effect on the function of microbial consortia.
In the past, several pathway models for the molecular mechanisms of microbial consortia were proposed. Please add these pathway models and introduce them briefly.
Please discuss the metabolic flux analysis for the molecular mechanisms of microbial consortia.
Response: We added several pathway models and discussed metabolic flux analysis for the molecular mechanisms of microbial consortia in section 3.2.2.
Please describe the novelty and innovation of this paper.
Response: In this article, we reviewed approaches of engineering microbial consortia especially for bioremediation, and highlight the several important principles.
Please describe the applicability.
Response: It could be used for the bioremediation of organic, metal, or combined pollutants.
Please discuss the limitations of practical application.
Please describe how to apply the results in environmental management.
Response: The limitations of practical application and ways of using engineering microbial consortia in environmental management have been added in section 5.
Reviewer 4 Report
To
The editor
It is my pleasure to review the manuscript titled “Engineering microbial consortia towards bioremediation” submitted to journal Water for publication. The manuscript is a review article on engineering microbial consortiums to enhance the bioremediation of pollutants. The review largely provides an overview of use of microbial consortiums with examples of few studies. The figures are simple drawings to depict the basic concepts. I did not find any major grammatical or spelling errors. My only critique of this review is that it is not detailed or analytical in nature. The conclusion should provide more detailed analysis of the information and authors should add future prospects on engineering microbial consortia
However, considering the scope of the journal I recommend to publish it after minor revision suggested above.
Thank you
Author Response
My only critique of this review is that it is not detailed or analytical in nature. The conclusion should provide more detailed analysis of the information and authors should add future prospects on engineering microbial consortia
Response: Thanks for your good words and useful suggestions on our manuscript.
We have rewritten the conclusion to provide more detailed analysis and added future prospects on engineering microbial consortia.
Round 2
Reviewer 1 Report
Thank you for taking the time to address some of my criticisms. Unfortunately, from my perspective most of the large problems I addressed at the top of my first review still remain. A large problem is that it isn’t clear why whole sections were chosen to review. As previously discussed, all of section 2 (cross-feeding) isn’t justified. As the authors themselves note on lines 33-36, species in consortia interact via many mechanisms, and these mechanisms change through time. In surveys of random microbial pairs, most interactions are negative (53%), and though positive interactions are common (~33%), very few interactions are mutualistic (~5%) (https://doi.org/10.1101/2020.06.24.169474 ). Since the manuscript is purportedly about “the molecular mechanisms of microbial consortia,” it isn’t clear why other common interaction mechanisms are not discussed.
A similar lack of justification occurs when deciding how to classify metabolites. The manuscript states that “It’s essential to identify the classification of metabolites as they have different influences on the stability of microbial consortia.” That makes sense to me. But then no explanation is given for why classifying via costly vs. costless helps us understand stability, and certainly there is no discussion why other methods of classifying (e.g. molecule type, connection to metabolism) would yield less valuable information. These sorts of missing explanations occur throughout.
From my perspective sections 3.1.1 and 3.1.2 really only apply to bottom-up, whereas 3.1.3 probably applies more to top-down. Yet, those sections are introduced as general considerations, and prior to introducing the concepts of bottom-up or top-down. Additionally, the abstract says that top-down and bottom-up engineering are a focus of the review, but only 5 short paragraphs are dedicated to these topics, which as a result barely get past defining what they are.
With one exception, the examples of successes in engineering for bioremediation do not connect back to the main thesis that understanding the molecular interactions of species in a consortia aids in engineering.
There are some very nice things in here, for example describing how the use of an accessory species (Bacillus) to increase bioavailability of a pollutant to the degrader (lines 172-180) is interesting, informative, and connects the engineering and ecological pieces together. Unfortunately, my position is that the manuscript as a whole could still benefit from the major changes I suggested in my initial revision.
Finally, there are minor, but present, language mistakes all throughout the manuscript. For example here are two mistakes in the introduction’s first paragraph:
1. “the interactions between microbial consortia” -- I believe the intent is to specify “the interactions [within] microbial consortia”, not to talk about consortium-consortium interactions.
2. “limited knowledge of the assembly and interaction of bacteria hinders” -- consortia assemble, but it is vague to think about “bacteria” assembling. Additionally, bacteria is plural, but interaction is singular.
Author Response
Thank you for taking the time to address some of my criticisms. Unfortunately, from my perspective most of the large problems I addressed at the top of my first review still remain. A large problem is that it isn’t clear why whole sections were chosen to review. As previously discussed, all of section 2 (cross-feeding) isn’t justified. As the authors themselves note on lines 33-36, species in consortia interact via many mechanisms, and these mechanisms change through time. In surveys of random microbial pairs, most interactions are negative (53%), and though positive interactions are common (~33%), very few interactions are mutualistic (~5%) (https://doi.org/10.1101/2020.06.24.169474 ). Since the manuscript is purportedly about “the molecular mechanisms of microbial consortia,” it isn’t clear why other common interaction mechanisms are not discussed.
Response: Many thanks for addressing these useful suggestions. We have added the discussions of interactions within microbial consortia (section 2.2). As you mentioned, the interactions within microbial consortia include antagonistic, neutral, or beneficial interactions. We thought when microbes associate with different species, the interactions are dynamic and may be influenced by environmental conditions. We also introduced the example of interactions changes under different environmental factors. And we hoped more researches needed to be done to shift interactions from negative to positive in the bioremediation of pollutants.
A similar lack of justification occurs when deciding how to classify metabolites. The manuscript states that “It’s essential to identify the classification of metabolites as they have different influences on the stability of microbial consortia.” That makes sense to me. But then no explanation is given for why classifying via costly vs. costless helps us understand stability, and certainly there is no discussion why other methods of classifying (e.g. molecule type, connection to metabolism) would yield less valuable information. These sorts of missing explanations occur throughout.
Response:
Thanks for your useful suggestions. Microbial cross-feedings are common in natural environment. Studies have reported that microbes significantly benefit from this exchange of metabolites (Harcombe, W.R.; Chacón, J.M.; Adamowicz, E.M.; Chubiz, L.M.; Marx, C.J. Evolution of Bidirectional Costly Mutualism from Byproduct Consumption, Proc. Natl. Acad. Sci., 2018). As you mentioned, these metabolites could be classified by molecule type, connection to metabolism and fitness cost. Recently, more researches tended to focus on costly and costless metabolites, as they have different influences on the stability of microbial consortia. For example, Bidirectional of costly metabolites may promote the growth of each species. In contrast, unidirectional exchange of costly metabolites harms the secretors, and may lead the microbial consortia to collapse. Besides, exchange of costless metabolites doesn’t harm the secretors, instead benefits the receivers, thus, it causes no effect on the stability of microbial consortia. So, in this article, secreted metabolites are mainly classified into two categories, costly metabolites and costless metabolites, depending on whether the secretion of metabolites causes a loss to the fitness of the organism. As you suggested, we have given more explanations for why classifying via costly vs. costless in our revised manuscript (section 2.1).
From my perspective sections 3.1.1 and 3.1.2 really only apply to bottom-up, whereas 3.1.3 probably applies more to top-down. Yet, those sections are introduced as general considerations, and prior to introducing the concepts of bottom-up or top-down. Additionally, the abstract says that top-down and bottom-up engineering are a focus of the review, but only 5 short paragraphs are dedicated to these topics, which as a result barely get past defining what they are.
Response:
Thanks for your useful suggestions. We have revised the whole section 3 as you suggested. We changed the orders of parts of section 3 and linked the specific principles to related methods of engineering, besides we expanded the paragraphs of top-down and bottom-up engineering. In detail, we firstly introduced the concepts, specific methods, and applications of top-down engineering, together with one important principle. We also discussed the advantages and disadvantages of it. Then, we addressed bottom-up engineering in the same way.
With one exception, the examples of successes in engineering for bioremediation do not connect back to the main thesis that understanding the molecular interactions of species in a consortia aids in engineering.
Response:
Thanks for your useful suggestions, we have revised this part to connect back to the main thesis.
There are some very nice things in here, for example describing how the use of an accessory species (Bacillus) to increase bioavailability of a pollutant to the degrader (lines 172-180) is interesting, informative, and connects the engineering and ecological pieces together. Unfortunately, my position is that the manuscript as a whole could still benefit from the major changes I suggested in my initial revision.
Response:
Many thanks for your good words for our manuscript. And we have followed your suggestions to revised our manuscript.
Finally, there are minor, but present, language mistakes all throughout the manuscript. For example here are two mistakes in the introduction’s first paragraph:
- “the interactions between microbial consortia” -- I believe the intent is to specify “the interactions [within] microbial consortia”, not to talk about consortium-consortium interactions.
Response: We have changed ‘between’ to ‘within’ in the whole revised manuscript.
- “limited knowledge of the assembly and interaction of bacteria hinders” -- consortia assemble, but it is vague to think about “bacteria” assembling. Additionally, bacteria is plural, but interaction is singular.
Response: We have revied this sentence (please see line 36).
Reviewer 3 Report
Acceptance is suggested.
Author Response
Response:
Many thanks for your good words for our manuscript.